# Extensive Dental Caries in Childhood: Association with Socioeconomic Status, Dietary and Daily Toothbrushing Frequency, and Sleep Disorders

**DOI:** 10.3390/ijerph23010043

**Published:** 2025-12-28

**Authors:** Patrícia Gomes Fonseca, Maria Letícia Ramos-Jorge, Jéssica Madeira Bittencourt, Karina Kendelhy Santos, Maria Eliza Consolação Soares, Priscilla Sena Souza Luz Campos, Cristiane Baccin Bendo, Izabella Barbosa Fernandes

**Affiliations:** 1Department of Dentistry, School of Biological and Health Sciences, Federal University of the Jequitinhonha and Mucuri Valleys, Diamantina, 39100000 MG, Brazil; patricia.fonseca@ufvjm.edu.br (P.G.F.); mlramosjorge@ufvjm.edu.br (M.L.R.-J.); karina.kendelhy@ufvjm.edu.br (K.K.S.); mariaeliza.soares@ufjf.br (M.E.C.S.); 2Department of Pediatric Dentistry, School of Dentistry, Federal University of Minas Gerais, Belo Horizonte, 31270901 MG, Brazil; jessbitten@ufmg.br (J.M.B.); priscillaluzcampos@ufmg.br (P.S.S.L.C.); izabellafernandes@ufmg.br (I.B.F.)

**Keywords:** dental caries, sleep disorders, sugars, child, toothbrushing

## Abstract

Dental caries is a prevalent childhood disease with a multifactorial etiology. The aim of the study is to evaluate the prevalence of extensive dental caries and its association with socioeconomic factors, dietary and daily toothbrushing frequency, and sleep disorders (SDs) in children aged 6 to 10 years. A cross-sectional study with 516 children and their caregivers was carried out. Socioeconomic information and data on dietary habits and oral health behaviors were obtained through a questionnaire administered to parents/caregivers. SDs were assessed using the Sleep Disturbance Scale for Children. Extensive dental caries was assessed using the ICDAS II (codes 5–6). Descriptive analyses and multivariate Poisson regression were performed. The prevalence of extensive dental caries was 20.7%. Extensive caries was associated with lower parental education (PR = 1.68; 95% CI: 1.16–2.44; *p* = 0.006), household income (PR = 5.64; 95% CI: 1.67–18.99; *p* = 0.005), frequent consumption of sugary snacks/drinks (PR = 2.74; 95% CI: 1.97–3.83; *p* < 0.001), and greater severity of SD (PR = 1.02; 95% CI: 1.00–1.03; *p* = 0.007). Extensive dental caries lesions are more common in children whose parents/caregivers have lower levels of education and income, consume more sugary foods/drinks, and have SDs.

## 1. Introduction

Dental caries, although showing a decrease in prevalence over time in Brazil, still represents a major public health challenge [1,2]. It is a prevalent disease among children with a multifactorial etiology, whose main causes include the presence of acidogenic bacterial biofilm [3], a diet rich in sugars, and inadequate oral hygiene practices [4]. In addition to these factors, dental caries is also influenced by socioeconomic determinants, which are often reflected in limited access to health services [5], as well as in diet quality and daily toothbrushing frequency [4].

The development of dental caries lesions is also modulated by various daily aspects and habits that affect children’s oral health [6]. Among these habits, sleep patterns (number of hours, quality, and regularity), chronotype (preference for specific sleep and wake times), and dietary habits stand out [6]. Evidence indicates that children with irregular sleep patterns are more likely to develop dental caries [6,7,8]. Disorders such as difficulty falling asleep, fragmented sleep, and irregular sleep schedules are associated with nighttime eating habits, especially the consumption of sugary foods and beverages, at a time when salivary flow is naturally reduced [6,7,8]. This combination favors cariogenic activity, contributing to the onset and progression of carious lesions [6].

Considering that the age range of 6 to 10 years represents a critical transitional phase in child development, marked by changes in dietary habits, hygiene behaviors, and sleep patterns, it becomes essential to understand how these factors interact and contribute to the risk of extensive dental caries.

By integrating socioeconomic, behavioral, and general health aspects, this study contributes to a more comprehensive understanding of the determinants of extensive dental caries in children. The combined analysis with sleep disorders highlights the complexity of this condition and reinforces the importance of interdisciplinary preventive approaches and public policies aimed at promoting child health. In this context, there is a clear need for integrated actions in school settings and primary care, focusing on health education, adequate nutrition, and sleep routines.

The aim of this study was to assess the prevalence of extensive dental caries and its association with socioeconomic factors, habits, and sleep disorders in children aged 6 to 10 years. The hypothesis of this study is that the prevalence of extensive dental caries in children aged 6 to 10 years is associated with sleep disorders, inadequate dietary and daily toothbrushing frequency, and those living in disadvantaged socioeconomic contexts.

## 2. Materials and Methods

### 2.1. Ethical Aspects and Study Setting

An epidemiological study was conducted in accordance with the ethical standards of the Brazilian National Health Council Resolution No. 466/12 [9] and was approved by the local Human Research Ethics Committee (Approval No. 5915760), following the recommendations of the “Strengthening the Reporting of Observational Studies in Epidemiology” (STROBE) guidelines for cross-sectional studies [10]. The parents or legal guardians of the children were invited to participate in the research, were informed about the study objectives, and those who agreed to the child’s participation signed the Informed Consent Form.

This study was carried out in the municipality of Diamantina, Minas Gerais, Brazil, which has 47,702 inhabitants and a Municipal Human Development Index (HDI) of 0.716 [11]. The city has eight public schools and four private schools in the urban area, and a total of 5832 children enrolled in elementary education (2387 children enrolled in elementary school level I), according to data from the Regional Education Office of Diamantina. The schooling rate for children aged 6 to 14 years was 97.8% in 2010 in Diamantina and its districts [11].

### 2.2. Study Design, Eligibility Criteria, and Sample Size Calculation

Children in the mixed dentition stage, aged 6 to 10 years, and their parents or legal guardians, enrolled in public and private schools in the municipality, participated in the study. Children with any neuropsychomotor disability reported by their caregivers were excluded from the study.

The sample size calculation was based on data from a pilot study, which reported a prevalence of extensive dental caries of 20.0%. Using a 99% confidence level, a 5% standard error and a correction factor of 1.2 to account for the cluster effect, a minimum sample of 433 children was calculated with the software “Open Source Epidemiologic Statistics for Public Health” (OpenEpi.com; version 3.01, accessed on 19 December 2025). A 20% increase was applied to account for potential sample losses, resulting in a final sample size of 520 children.

Using a stratified sampling technique, the study sample was selected considering the twelve schools (public and private) in the municipality as sampling units. First, an electronic lottery was carried out to select the participating schools by assigning random numbers to each school, respecting the proportion of public and private schools in the municipality. Thus, four public schools and two private schools were randomly selected for the study. Subsequently, the children were selected through a random draw from the enrollment lists, considering the age range of 6 to 10 years.

### 2.3. Training, Calibration, and Pilot Study

A theoretical–practical training was conducted for the diagnosis of dental caries using images representing different stages of caries lesion progression, based on the International Caries Detection and Assessment System—ICDAS-II [12]. Subsequently, a clinical dental examination of 10% of the total study sample was performed for examiner calibration. The children were recruited from the Pediatric Dentistry Clinics of the Federal University of the Jequitinhonha and Mucuri Valleys (UFVJM), where they received dental care. The oral clinical examination of the children was carried out by the examiner and an experienced researcher to verify inter-examiner agreement. After a one-week interval, the children were examined again to verify intra-examiner agreement. Good intra- (Kappa coefficient > 0.88) and inter-examiner (Kappa coefficient > 0.76) agreement was observed for the assignment of ICDAS-II codes related to dental caries. Agreement for the assessment of extensive dental caries (ICDAS codes 5 and 6) was also evaluated, showing excellent intra-examiner (Kappa coefficient = 0.96) and inter-examiner (Kappa coefficient = 0.94) reliability.

A total of 30 children and their caregivers participated in the pilot study to test the methodology to be applied in the main study, including the administration of questionnaires and the clinical dental examination for the detection of dental caries. As no need for methodological adjustments was identified, the children from the pilot study were included in the final sample.

### 2.4. Data Collection: Non-Clinical Variables

To collect information about the child and family, a questionnaire was sent to the parents or legal guardians, covering general child data (sex and age); socioeconomic characteristics of the family: monthly household income (greater than two minimum wages or less than or equal to two minimum wages), number of income dependents (up to three or more than three people), and the highest educational level among the parents or guardians (eight or more years of schooling or less than eight years of schooling). Information regarding the child’s oral hygiene practices was also collected through questions on toothbrushing frequency (two or more times per day or less than twice per day) and the frequency of consumption of foods/beverages with added sugar (greater than twice per day or fewer than twice per day). In addition, sleep disturbances were assessed using the “Sleep Disturbance Scale for Children” (SDSC) [13]. The SDSC is based on a Likert scale, and the total score corresponds to the sum of the responses to each item, according to the frequency with which symptoms occur: “Never,” “Occasionally” (1–2 times per month), “Sometimes” (1–2 times per week), “Often” (3–5 times per week), and “Always” (every day) [13]. The total SDSC score may range from 26 to 130 points, with higher scores indicating greater severity of sleep disturbances in the child [12]. The SDSC includes questions related to disorders of initiating and maintaining sleep, sleep-disordered breathing, arousal disorders, excessive somnolence, sleep hyperhidrosis, and sleep–wake transition disorders, considering the previous six months [13].

### 2.5. Data Collection: Clinical Variable

A clinical dental examination of children was performed in their respective schools using artificial light (headlamp) to detect dental caries, based on the ICDAS-II [12]. During the caries assessment, each tooth surface was examined and classified using codes 0 to 6, according to the progression of the caries process. Dental caries severity was assessed using the International Caries Detection and Assessment System (ICDAS) and managed according to the International Caries Classification and Management System (ICCMS). In the present study, extensive dental caries was defined as cavitated lesions classified as ICDAS codes 5 and 6, which correspond to distinct cavities with visible dentin and extensive distinct cavities with visible dentin. These criteria are aligned with the ICCMS clinical framework (https://www.iccms-web.com/, (accessed on 19 December 2025). Children presenting at least one dental caries lesion classified as ICDAS-II code 5 or 6, in either primary or permanent teeth, were classified as having the presence of extensive dental caries. All dental examinations were performed by a previously calibrated examiner.

### 2.6. Statistical Analysis

Data were entered/coded and analyzed using the IBM SPSS Statistics (SPSS for Windows, version 25.0, IBM Inc., Armonk, NY, USA) and Stata software (version 16.1, Stata Corp., College Station, TX, USA). Descriptive analyses were performed to characterize the sample. The main dependent variable of the study was the presence of extensive dental caries (ICDAS-II codes 5–6). The Mann–Whitney test was used to compare the distribution of mean scores on the SDSC items according to the presence of extensive dental caries.

Unadjusted Poisson regression analysis with robust variance was first used to estimate associations between each independent variable and extensive dental caries. Independent variables with a significance level of up to 20% (*p* < 0.20) in the unadjusted analyses, as well as variables considered relevant to the theoretical model [6,14,15,16], were retained for adjustment in the multiple regressions.

Adjusted multiple Poisson regression models with robust variance were conducted. All analyses were conducted taking into account the weights of the type of school of each participant, since a multistage sampling method was used instead of simple random sampling. Variables were organized into hierarchical levels, from distal to proximal determinants. The levels comprised sociodemographic factors (parental education level, household income, and number of dependents on the monthly household income), child characteristics (sex and age), and child habits (consumption of sugary snacks/drinks, daily toothbrushing frequency, and sleep disorders assessed using the SDSC score), in this order. Independent variables with a *p*-value < 0.05 after adjustment for variables at the same or preceding hierarchical levels were retained in the final models. Prevalence ratios (PR) were calculated with their 95% confidence intervals (95% CI), and statistical significance was set at *p* < 0.05.

## 3. Results

The study included 516 children. Sample loss corresponded to 8.3%, resulting from incomplete questionnaire responses provided by caregivers. The prevalence of extensive dental caries (ICDAS-II codes 5–6) in the total sample was 20.7%. A predominance of children aged 9 to 10 years (52.7%) and females (50.78%) was observed.

Table 1 presents the characteristics of the study sample, distributed according to the presence of extensive dental caries, and the unadjusted analysis. The total SDSC score was higher among children with extensive dental caries (Mean = 42.16; SD = 13.22; *p* = 0.008). The prevalence of extensive dental caries was higher among children from families with a monthly household income of two minimum wages or less (22.3%; *p* = 0.025), whose parents or guardians had fewer than eight years of schooling (31.8%; *p* = 0.003), and who consumed sugary snacks/beverages more than twice per day (38,1%; *p* < 0.001).

In the hierarchical multiple analysis (Table 2), the results of the first level demonstrated an association of extensive dental caries with parental education level (*p* = 0.016) and household income (*p* = 0.003). On the second level, adjusted by the variables of the previous level, no significant associations were found between extensive dental caries and child characteristics (*p* > 0.05). Thus, in the third level, extensive dental caries was associated with the consumption of sugary snacks/beverages (*p* < 0.001) and sleep disturbances (*p* = 0.033).

The final adjusted multiple model (Table 2) revealed a significant association between the presence of extensive dental caries and the following factors: parental or guardian educational level below eight years of schooling (PR = 1.68; 95% CI: 1.16–2.44; *p* = 0.006); lower monthly household income (PR = 5.64; 95% CI: 1.67–18.99; *p* = 0.005), consumption of sugary snacks/beverages more than twice per day (PR = 2.74; 95% CI: 1.97–3.83; *p* < 0.001); and total SDSC score (PR = 1.02; 95% CI: 1.00–1.03; *p* = 0.007).

Table 3 presents the distribution of mean (SD) scores for SDSC items according to the presence of extensive dental caries. Children with extensive dental caries took longer to fall asleep (mean = 1.97; SD = 1.18; *p* = 0.033), more frequently reported difficulty falling asleep (mean = 1.67; SD = 1.14; *p* = 0.041), and moved continuously during sleep (mean = 2.54; SD = 1.48; *p* = 0.011) compared with children without extensive dental caries.

## 4. Discussion

This study aimed to investigate the prevalence of extensive dental caries in schoolchildren, as well as its association with socioeconomic factors, habits, and sleep disturbances. A considerable proportion of the children presented extensive dental caries, and this condition was associated with lower parental or caregiver educational level, frequent consumption of sugary foods and beverages, and the presence of sleep disturbances. Therefore, the findings of this study are consistent with the proposed hypothesis.

The high prevalence of extensive dental caries observed in this study, classified as codes 5 and 6 according to the ICDAS II system, reflects the severity of the condition, suggesting, in most cases, the need for invasive treatments, as these codes correspond to the most advanced stages of the disease [5]. A study with Brazilian schoolchildren in the same age range (6 to 10 years) identified an even higher prevalence of untreated dental caries (58.6%), assessed using the Decayed, Missing and Filled Teeth (DMFT/dmft) index, indicating a high proportion of untreated cases [17]. Such discrepancies may reflect methodological, regional, socioeconomic differences, as well as variations in public oral health policies.

There was an association between the presence of extensive dental caries in children and lower parental or caregiver education (less than eight years of schooling). Previous studies indicate that parents’ educational level influences the occurrence of dental caries in children [14,18,19]. In general, parents with higher education tend to adopt more adequate oral hygiene practices and to seek preventive dental care for their children, which contributes to a lower prevalence of caries [14].

In the present study, even after statistical adjustment, the association between household income and dental caries remained significant. In general, monthly household income can influence children’s eating patterns, favoring cariogenic diets and, consequently, increasing the risk of developing dental caries [15]. In addition, unfavorable socioeconomic conditions hinder access to dental services, especially among low-income families [20]. Barriers such as financial limitations, lack of flexibility at work, and precarious living conditions compromise the prevention and treatment of oral diseases, making access difficult even when services are free or low-cost [20].

Although dietary patterns vary within the study population itself, this research identified an association between the frequent consumption of sugary foods and beverages (more than twice a day) [21] and the presence of extensive dental caries in children. This association may be interpreted in light of biological mechanisms described in the literature, as regular sugar intake—especially sucrose—has been shown to favor the development of carious lesions by serving as a substrate for acidogenic bacteria such as Streptococcus mutans, which produce acids responsible for enamel demineralization [22]. These results are consistent with previous studies, which reported a higher prevalence and progression of dental caries in children with a high frequency of sweets and sugar consumption [4,15].

Literature reports that sleep problems may contribute to the development of dental caries in children [6,7,8,23,24,25]. These findings are consistent with the results of the present study, which observed a higher SDSC total score among children with extensive dental caries. Using the same sleep disturbances instrument as the present study, a previous study found that higher total SDSC scores were associated with untreated dental caries in children [26]. Our findings complement these results by demonstrating that specific SDSC items related to sleep onset and sleep fragmentation (taking longer to fall asleep, difficulty falling asleep and continuous movements during sleep) were significantly different between children with and without extensive dental caries. These findings suggest that sleep initiation difficulties and sleep fragmentation were the sleep-related aspects most strongly associated with extensive dental caries in this population. Comparable patterns have been reported in previous studies, although assessed using different sleep questionnaires [8,23,24,27,28]. Mehdipour et al. (2024) demonstrated that dental caries at different stages of progression, also assessed using the ICDAS, was associated with inadequate sleep patterns in children aged 8–12 years, especially nocturnal awakenings [28]. A population-based study of Korean children aged 6–12 years reported that short sleep duration and sleep-related behaviors were associated with poorer oral health indicators, including dental caries [23]. A systematic review, although addressing children in a different age group, demonstrated that irregular or late bedtimes and shorter sleep duration may be risk factors for dental caries [8]. Although these studies did not focus on specific symptoms of sleep initiation or maintenance, together with our findings, they suggest that irregular sleep patterns are relevant to children’s oral health.

Difficulties in falling asleep and restless sleep may reflect irregular nighttime routines and unhealthy behaviors, such as increased nocturnal sugar consumption and inadequate oral hygiene, as well as potential physiological changes related to salivary flow [6,28]. A review suggests that children with irregular sleep schedules or an evening chronotype (characterized by later bedtimes and wake times) are at greater risk of caries because they tend to consume sugary foods and drinks at night, particularly before bedtime [6]. Although salivary flow was not assessed in the present study, the literature suggests that nighttime sugar consumption coincides with a natural reduction in salivary flow during sleep, which is an important protective factor against dental caries [6,29]. However, this pathway was not evaluated in this study. Specifically, the consumption of sugary foods or beverages immediately before bedtime was not assessed, which represents a limitation of the present study. Given the nature of the data collected, it was not possible to directly observe or measure these behaviors. Saliva plays a crucial role in neutralizing acids produced by oral bacteria from dietary sugars and in supporting enamel remineralization [6]. Thus, the combination of nighttime sugar intake and decreased salivary production has been described in the literature as a factor associated with an increased risk of dental caries [6,7].

According to the results obtained, no association was observed between extensive dental caries in children and toothbrushing frequency. Generally, a higher frequency of toothbrushing is expected to be associated with a lower prevalence of dental caries [15,30]. However, this lack of association may be explained by the limitations of the measure used, as daily toothbrushing frequency alone is insufficient to capture the true protective effect of oral hygiene. Important aspects such as brushing quality, technique, duration, and caregiver supervision were not assessed in this study and may play a critical role in caries prevention [31].

Among the limitations of the present study, the absence of an assessment of the consumption of sugary foods and drinks immediately before bedtime stands out. This is a critical period for the development of dental caries due to the reduction in salivary flow and its frequent association with unhealthy habits, such as less frequent toothbrushing [6]. Considering this variable could have provided a better understanding of the worsening of caries lesions in the population analyzed.

Another limitation of this study is the absence of information on other well-established caries-related factors described in the literature, such as caregivers’ caries status, the age at which oral hygiene practices were initiated, and fluoride exposure, including the use of fluoride toothpaste. These variables were not assessed in the parent questionnaire and, therefore, could not be included in the adjusted models, which may limit a more comprehensive interpretation of the associations observed.

Finally, it is important to note that, as the data of the present study were obtained from a single municipality in southeastern Brazil, the findings may have limited external validity and should be interpreted with caution when extrapolated to other settings or contexts. Regional differences in socioeconomic conditions, access to dental care, and health-related behaviors may influence the occurrence of sleep disorders and dental caries. Therefore, further studies conducted in diverse settings are needed to confirm and expand upon these findings.

Future research may benefit from longitudinal designs and intervention studies that explore the factors associated with extensive dental caries identified in this study. In this context, emphasis on educational actions focused on sleep hygiene and reducing the consumption of sugary foods and beverages, combined with expanded access to preventive dental care, is particularly relevant. Such initiatives may contribute to effective strategies for preventing and managing caries, especially among children exposed to greater vulnerability, such as those with caregivers who have lower levels of education and those experiencing sleep disturbances.

## 5. Conclusions

The findings of the present study indicated that extensive dental caries was associated with factors such as lower parental or caregiver educational level, household income, frequent consumption of sugary foods and beverages, and the presence of sleep disturbances. The combined analysis of these variables contributes to understanding modifiable behavioral factors, such as dietary and sleep habits, which can support the development of strategies aimed at preventing and managing dental caries in childhood.

## Figures and Tables

**Table 1 ijerph-23-00043-t001:** Descriptive analysis and unadjusted Poisson regression analysis for independent variables and extensive dental caries (*n* = 516).

Independent Variables	Extensive Dental Caries (ICDAS-II Codes 5–6)				
	No *n* (%)	Yes *n* (%)	Total *n* (%)	Unadjusted PR 95% CI	*p*-Value
Highest parental education level *					
≥8 years of schooling	345 (81.9)	76 (18.1)	421 (100)	1	
<8 years of schooling	58 (68.2)	27 (31.8)	85 (100)	1.76 (1.21–2.55)	**0.003**
Household income					
>2 minimum wages	46 (93.9)	3 (6.1)	49 (100)	1	
≤2 minimum wages	363 (77.7)	104 (22.3)	467 (100)	3.57 (1.18–10.84)	**0.025**
Number of dependents on monthly household income					
0 to 3 people	159 (84.1)	30 (15.9)	189 (100)	1	
More than 3 people	250 (76.5)	77 (23.5)	327 (100)	1.47 (1.00–2.17)	0.052
Sex					
Female	213 (81.3)	49 (18.7)	262 (100)	1	
Male	196 (77.2)	58 (22.8)	254 (100)	1.26 (0.89–1.78)	0.194
Age					
6 to 8 years	185 (75.8)	59 (24.2)	244 (100)	1	
9 to 10 years	224 (82.4)	48 (17.6)	272 (100)	0.73 (0.52–1.04)	0.081
Consumption of snacks/sugary drinks > 2 times per day					
No	323 (85.7)	54 (14.3)	377 (100)	1	
Yes	86 (61.9)	53 (38.1)	139 (100)	2.78 (1.99–3.87)	**<0.001**
Daily toothbrushing frequency					
2 times or more per day	315 (80.4)	77 (19.6)	392 (100)	1	
<2 times per day	94 (75.8)	30 (24.2)	124 (100)	1.17 (0.79–1.72)	0.432
Total SDSC score—Mean (SD)	39.15 (9.83)	42.16 (13.22)	39.78 (10.68)	1.02 (1.00–1.03)	**0.008**

SDSC: Sleep Disturbance Scale for Children; SD = Standard Deviation; Prevalence Ratio (PR); CI: Confidence Interval; Values in bold type are statistically significant (*p* < 0.05); * There are 10 missing values for the parental education variable.

**Table 2 ijerph-23-00043-t002:** Adjusted Poisson regression analysis for independent variables and extensive dental caries (*n* = 516).

Variables	Hierarchical Model Adjusted PR 95% CI	*p*-Value	Final Model Adjusted PR 95% CI	*p*-Value
**1st level—sociodemographic factors**				
Highest parental education level				
≥8 years of schooling	1.00		1.00	
<8 years of schooling	1.67 (1.15–2.43)	**0.016**	1.68 (1.16–2.44)	**0.006**
Household income				
>2 minimum wages	1.00		1.00	
≤2 minimum wages	1.84 (1.06–19.75)	**0.003**	5.64 (1.67–18.99)	**0.005**
Number of dependents on monthly household income				
0 to 3 people	1.00			
More than 3 people	1.37 (0.92–2.04)	0.124	-	-
**2nd level—child characteristics**				
Sex				
Female	1.00			
Male	1.19 (0.884–1.70)	0.318	-	-
Age				
6 to 8 years	1.00			
9 to 10 years	0.79 (0.55–1.12)	0.186	-	-
**3rd level—child habits**				
Consumption of snacks/sugary drinks > 2 times per day				
No	1.00		1.00	
Yes	2.77 (1.94–3.96)	**<0.001**	2.74 (1.97–3.83)	**<0.001**
Total SDSC score—Mean (SD)	1.01 (1.00–1.03)	**0.033**	1.02 (1.00–1.03)	**0.007**

SDSC: Sleep Disturbance Scale for Children; Prevalence Ratio (PR); CI: Confidence Interval; Values in bold type are statistically significant (*p* < 0.05).

**Table 3 ijerph-23-00043-t003:** Distribution of mean scores on the Sleep Disturbance for Children (SDSC) items according to the presence of extensive dental caries (*n* = 516).

SDSC Items	Extensive Dental Caries		
	No Mean (SD)	Yes Mean (SD)	*p*-Value
How many hours does the child sleep per night?	1.76 (0.80)	1.71 (0.84)	0.429
2. How long does it take the child to fall asleep?	1.67 (0.89)	1.97 (1.18)	**0.033**
3. The child does not want to go to bed to sleep.	1.90 (1.09)	1.99 (1.19)	0.574
4. The child has difficulty falling asleep.	1.41 (0.79)	1.67 (1.14)	**0.041**
5. Before falling asleep, the child is restless, nervous, or afraid.	1.48 (0.90)	1.59 (1.04)	0.554
6. The child shows sudden movements, jerks, or tremors when falling asleep.	1.31 (0.76)	1.59 (1.17)	0.053
7. During the night, the child makes rhythmic movements with the head and neck.	1.35 (0.83)	1.59 (1.18)	0.090
8. The child says that he/she sees strange things shortly before falling asleep.	1.19 (0.58)	1.25 (0.69)	0.425
9. The child sweats excessively when falling asleep.	1.62 (1.09)	1.75 (1.17)	0.221
10. The child wakes up more than twice during the night.	1.34 (0.66)	1.50 (0.99)	0.575
11. The child wakes up during the night and has difficulty falling asleep again.	1.33 (0.70)	1.42 (0.92)	0.780
12. The child moves continuously during sleep.	2.15 (1.33)	2.54 (1.48)	**0.011**
13. The child does not breathe well during sleep.	1.40 (0.89)	1.51 (1.11)	0.971
14. The child stops breathing during sleep.	1.10 (0.44)	1.20 (0.71)	0.217
15. The child snores.	1.84 (1.21)	2.06 (1.45)	0.385
16. The child sweats excessively during the night.	1.58 (1.07)	1.80 (1.26)	0.110
17. The child gets up, sits up in bed, or walks while sleeping.	1.34 (0.78)	1.31 (0.78)	0.439
18. The child talks during sleep.	1.70 (0.98)	1.87 (1.15)	0.257
19. The child grinds his/her teeth during sleep.	1.71 (1.20)	1.63 (1.09)	0.591
20. During sleep, the child screams in distress without waking up.	1.18 (0.52)	1.25 (0.77)	0.679
21. The child has nightmares that he/she does not remember the next day.	1.41 (0.74)	1.42 (0.75)	0.880
22. The child has difficulty waking up in the morning.	1.91 (1.21)	1.84 (1.28)	0.206
23. The child wakes up feeling tired in the morning.	1.53 (0.99)	1.55 (1.07)	0.878
24. Upon awakening, the child is unable to move or feels paralyzed for several minutes.	1.22 (0.62)	1.18 (0.68)	0.109
25. The child feels sleepy during the day.	1.44 (0.75)	1.55 (1.04)	0.903
26. During the day, the child falls asleep unexpectedly, without warning.	1.27 (0.61)	1.42 (0.92)	0.285

Mann–Whitney test; SDSC: Sleep Disturbance Scale for Children; SD: standard deviation; Prevalence Ratio (PR); CI: Confidence Interval; Values in bold type are statistically significant (*p* < 0.05).

## Data Availability

The raw data supporting the conclusions of this article will be made available by the authors on request.

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
