# Peer review of "Int. J. Environ. Res. Public Health2026, 23(1), 43;https://doi.org/10.3390/ijerph23010043"

_ijerph, 2025, doi:10.3390/ijerph23010043_

Round 1

Reviewer 1 Report

Comments and Suggestions for Authors

Dear Authors,
Thank you for the opportunity to review your manuscript. Several elements may benefit from clarification to strengthen methodological consistency and the interpretation of the findings. The enclosed comments are intended to support further refinement of the manuscript.

Author Response

Reviewer #1

TITLE

Reviewer’s comment: The title is generally appropriate and reflects the main concepts addressed in the study. However, it may be helpful to clarify within the manuscript how the term “extensive dental caries” is defined, as this terminology varies across diagnostic frameworks. In addition, the expression “oral hygiene habits” encompasses a broader range of behaviors, whereas the study specifically evaluates toothbrushing frequency. Clarifying this distinction in the text may improve precision and alignment between the title and the variables assessed. These are minor considerations that do not affect the scientific validity of the manuscript.

Authors’ response: We thank the reviewer for this valuable comment. In the revised version of the manuscript, we clarified that extensive dental caries was defined according to the International Caries Detection and Assessment System (ICDAS), considering codes 5 and 6 as indicative of extensive carious lesions. This definition is now explicitly stated in the Methods section, with appropriate reference to the ICDAS criteria:

“Dental caries severity was assessed using the International Caries Detection and Assessment System (ICDAS) and managed according to the International Caries Classification and Management System (ICCMS). In the present study, extensive dental caries was defined as cavitated lesions classified as ICDAS codes 5 and 6, which correspond to distinct cavities with visible dentin and extensive distinct cavity with visible dentin. These criteria are aligned with the ICCMS clinical framework (https://www.iccms-web.com/).” (line 157)

Additionally, throughout the manuscript, the term “oral hygiene habits” was replaced by terminology that more accurately reflects the variables assessed, specifically “daily toothbrushing frequency.” These changes were made to improve clarity.

Reviewer’s comment: The introduction is generally well structured and adequately supported by recente literature. One point to revise is the formulation of the study hypothesis. As the design is cross-sectional, causal wording (“is higher among children who…”) may not be appropriate. Rephrasing the hypothesis to reflect association rather than causality would improve methodological alignment.

Authors’ response: We thank the reviewer for this important methodological observation. As suggested, the study hypothesis was revised to avoid causal wording and to reflect an associative relationship, which is more appropriate for a cross-sectional design. The revised hypothesis now emphasizes associations rather than causality, ensuring better methodological alignment without altering the objectives of the study.

“The hypothesis of this study is that the prevalence of extensive dental caries in children aged 6 to 10 years is associated with sleep disorders, inadequate dietary habits, lower daily toothbrushing frequency, and disadvantaged socioeconomic contexts.” (line 75)

MATERIALS AND METHODS

Reviewer’s comment: 1. Sample size calculation The sample size was calculated based on the prevalence of untreated caries (58.6%), whereas the primary outcome analyzed in this study was extensive dental caries (ICDAS 5–6), which presents substantially lower prevalence. Using a prevalence estimate that does not correspond to the primary outcome may affect the theoretical adequacy of the sample size calculation. Clarifying the rationale for this approach, or aligning the estimate with the primary outcome, would improve methodological transparency.

Authors’ response: We appreciate the reviewer’s comment. The sample size calculation was revised based on data from the pilot study, as follows:

“The sample size calculation was based on data from a pilot study, which reported a prevalence of extensive dental caries of 20.0%. Using a 99% confidence level, a 5% standard error and a correction factor of 1.2 to account for the cluster effect, a minimum sample of 433 children was calculated with the software “Open Source Epidemiologic Statistics for Public Health” (OpenEpi.com). A 20% increase was applied to account for potential sample losses, resulting in a final sample size of 520 children." (line 100)

Reviewer’s comment: 2. Variable selection for the multivariable model. In the statistical analysis section, the criteria used for selecting variables for the final adjusted Poisson regression model are not described. Providing the inclusion criterion (e.g., theoretical relevance, p-value threshold from univariate analysis) would enhance the reproducibility of the analytical process.

Authors’ response: We appreciate the reviewer’s comment. The Statistical Analysis section has been revised to improve clarity regarding the criteria used for variable selection in the final adjusted Poisson regression model:

“Unadjusted Poisson regression analysis with robust variance was first used to estimate associations between each independent variable and extensive dental caries. Independent variables with a significance level of up to 20% (p < 0.20) in the unadjusted analyses, as well as variables considered relevant to the theoretical model [6,13–15], were retained for adjustment in the multiple regressions.” (line 175)

“Adjusted multiple Poisson regression models with robust variance were conducted. All analyses were conducted taking into account the weights of the type of school of each participant, since multistage sampling method was used instead of simple random sampling. Variables were organized into hierarchical levels, from distal to proximal determinants. The levels comprised sociodemographic factors (parental education level, household income, and number of dependents on the monthly household income), child characteristics (sex and age), and child habits (consumption of sugary snacks/drinks, daily toothbrushing frequency, and sleep disorders assessed using the SDSC score), in this order. Independent variables with a p-value < 0.05 after adjustment for variables at the same or preceding hierarchical levels were retained in the final models. Prevalence ratios (PR) were calculated with their 95% confidence intervals (95% CI), and statistical significance was set at p < 0.05.” (line 180)

RESULTS

Reviewer’s comment: Selection of variables for the multivariable model The manuscript reports the final adjusted Poisson regression model; however, the criteria used to determine which variables were retained or excluded from the model are not described. Providing a brief explanation of the selection strategy (e.g., theoretical relevance, predefined criteria, or thresholds from univariate analysis) would enhance the transparency and reproducibility of the analytical process.

Authors’ response: We appreciate the reviewer’s comment. The Statistical Analysis section has been revised to improve clarity regarding the criteria used for variable selection in the final adjusted Poisson regression model:

“Unadjusted Poisson regression analysis with robust variance was first used to estimate associations between each independent variable and extensive dental caries. Independent variables with a significance level of up to 20% (p < 0.20) in the unadjusted analyses, as well as variables considered relevant to the theoretical model [6,13–15], were retained for adjustment in the multiple regressions.” (line 175)

“Adjusted multiple Poisson regression models with robust variance were conducted. All analyses were conducted taking into account the weights of the type of school of each participant, since multistage sampling method was used instead of simple random sampling. Variables were organized into hierarchical levels, from distal to proximal determinants. The levels comprised sociodemographic factors (parental education level, household income, and number of dependents on the monthly household income), child characteristics (sex and age), and child habits (consumption of sugary snacks/drinks, daily toothbrushing frequency, and sleep disorders assessed using the SDSC score), in this order. Independent variables with a p-value < 0.05 after adjustment for variables at the same or preceding hierarchical levels were retained in the final models. Prevalence ratios (PR) were calculated with their 95% confidence intervals (95% CI), and statistical significance was set at p < 0.05.” (line 180)

DISCUSSION

Reviewer’s comment: 1. Interpretation and causal wording Some statements in the Discussion (e.g., “the study hypothesis was confirmed”) could imply causal interpretation, which is not supported by the cross-sectional design. Rephrasing these statements to reflect association rather than causality would improve methodological alignment.

Authors’ response: We thank the reviewer for this important comment. As suggested, statements in the Discussion that could imply causal interpretation were revised to reflect associative relationships, which are more appropriate for the cross-sectional design of the study. Expressions such as “the study hypothesis was confirmed” were rephrased to avoid causal inference, ensuring better methodological alignment without altering the interpretation of the findings.

Reviewer’s comment: 2. Explanation of collinearity assumption The manuscript suggests that the loss of significance for household income in the adjusted model may be due to collinearity with parental education. As no collinearity diagnostics are reported, clarifying whether this was formally assessed would enhance the transparency of the analytical interpretation.

Authors’ response: We performed a new multiple regression analysis to obtain an updated adjusted model, incorporating the sampling weights related to the type of school for each participant, as a multistage sampling design was used rather than simple random sampling. In this weighted adjusted model, household income remained statistically significant. The revised results are presented in Table 2, and the previous discussion about collinearity was removed from the Discussion section.

“In the present study, even after statistical adjustment, the association between household income and dental caries remained significant.” (line 265)

Reviewer 2 Report

Comments and Suggestions for Authors

It is mentioned in the statistical analysis plan that the final adjustment of the model will retain "variables that are important for explaining the theoretical framework". The research emphasizes the multifactorial nature of dental caries and identifies key risk factors such as the caries status of caregivers, the age at which oral hygiene began, and a history of fluoride exposure (such as the use of fluoride toothpaste) (PMID: 36330110, 36122565). Did your parent questionnaire include the measurement of these variables? If not, how will this affect your explanation of the association between sleep disorders and widespread dental caries? Sleep disorders were evaluated using the total score (continuous variable) of the SDSC scale. However, research has found that the association between sleep duration (such as "short sleep time") as a categorical variable and the risk of dental caries may be more definite (PMID: 41153581). In addition to using the total score, do you plan to classify the SDSC score or specific dimensions within it (such as "sleep onset and maintenance disorders") (for example, by critical values as "sleep disorders with/without clinical significance") to examine the dose-response relationship between sleep problems of different severity levels and the risk of dental caries? The Kappa values between and within the examiners after calibration were reported to be > 0.76, indicating good consistency. To achieve the rigor of similar high-standard research in the [Knowledge Base] (PMID: 36122565), could you provide more detailed calibration results? For example, what are the Kappa values for the specific thresholds (ICDAS-II codes 5 and 6) used to define "generalized caries"? This is crucial for ensuring the measurement accuracy of the result variable. It is an advantage that random sampling stratified by school type (public/private) was adopted. Given that your research hypothesis involves the key variable of socioeconomic status (SES), and the type of school (public vs. Private is usually a strong proxy metric for SES. In the subsequent analysis, do you plan to incorporate "school type" as a stratified variable or covariate into the model to more clearly dissect the possible complex relationship between SES, sleep habits and dental caries?

Author Response

Reviewer #2

Reviewer’s comment: It is mentioned in the statistical analysis plan that the final adjustment of the model will retain "variables that are important for explaining the theoretical framework". The research emphasizes the multifactorial nature of dental caries and identifies key risk factors such as the caries status of caregivers, the age at which oral hygiene began, and a history of fluoride exposure (such as the use of fluoride toothpaste) (PMID: 36330110, 36122565). Did your parent questionnaire include the measurement of these variables? If not, how will this affect your explanation of the association between sleep disorders and widespread dental caries?

Authors’ response: We thank the reviewer for this important observation. The parent questionnaire did not include information on caregivers’ caries status, age at initiation of oral hygiene practices, or fluoride exposure (such as the use of fluoride toothpaste). Therefore, these variables were not included in the analytical models.

We acknowledge that the absence of these well-established caries-related factors represents a limitation of the present study which may limit a more comprehensive model of dental caries in childhood. This limitation has now been explicitly addressed in the Discussion section.

“Another limitation of this study is the absence of information on other well-established caries-related factors described in the literature, such as caregivers’ caries status, the age at which oral hygiene practices were initiated, and fluoride exposure, including the use of fluoride toothpaste. These variables were not assessed in the parent questionnaire and, therefore, could not be included in the adjusted models, which may limit a more comprehensive interpretation of the associations observed.” (line 336)

Reviewer’s comment: Sleep disorders were evaluated using the total score (continuous variable) of the SDSC scale. However, research has found that the association between sleep duration (such as "short sleep time") as a categorical variable and the risk of dental caries may be more definite (PMID: 41153581). In addition to using the total score, do you plan to classify the SDSC score or specific dimensions within it (such as "sleep onset and maintenance disorders") (for example, by critical values as "sleep disorders with/without clinical significance") to examine the dose-response relationship between sleep problems of different severity levels and the risk of dental caries?

Authors’ response: We appreciate the reviewer’s comment. Instead of using only the total score, and to examine the association of each SDSC item with extensive dental caries, we included a new table (Table 3) presenting the distribution of mean (SD) scores for SDSC items according to the presence of extensive dental caries. Accordingly, modifications were made to the Statistical Analysis, Results, and Discussion sections.

“The Mann–Whitney test was used to compare the distribution of mean scores on the SDSC items according to the presence of extensive dental caries.” (line 172)

 “Table 3 presents the distribution of mean (SD) scores for SDSC items according to the presence of extensive dental caries. Children with extensive dental caries took longer to fall asleep (mean = 1.97; SD = 1.18; p = 0.033), more frequently reported difficulty falling asleep (mean = 1.67; SD = 1.14; p = 0.041), and moved continuously during sleep (mean = 2.54; SD = 1.48; p = 0.011) compared with children without extensive dental caries.” (line 217)

 “An analysis for each item of the SDSC demonstrated that children with extensive dental caries took longer to fall asleep and more frequently reported difficulty falling asleep, as well as moved continuously during sleep, compared with children without extensive dental caries.” (line 255)

 “Literature reports that sleep problems may contribute to the development of dental caries in children [6–8, 22-24]. These findings are consistent with the results of the present study, which observed higher SDSC total score among children with extensive dental caries. Using the same sleep disturbances’ instrument as the present study, a previous study found that higher total SDSC scores were associated with untreated dental caries in children [25]. Our findings complement these results by demonstrating that specific SDSC items related to sleep onset and sleep fragmentation (taking longer to fall asleep, difficulty falling asleep and continuous movements during sleep) were significantly different between children with and without extensive dental caries. These findings suggest that sleep initiation difficulties and sleep fragmentation were the sleep-related aspects most strongly associated with extensive dental caries in this population. Comparable patterns have been reported in previous studies, although assessed using different sleep questionnaires [8,22-23,26,27]. Mehdipour et al. (2024) demonstrated that dental caries at different stages of progression, also assessed using the ICDAS, was associated with inadequate sleep patterns in children aged 8–12 years, especially nocturnal awakenings [27]. A population-based study of Korean children aged 6–12 years reported that short sleep duration and sleep-related behaviors were associated with poorer oral health indicators, including dental caries [22]. A systematic review, although addressing children in a different age group, demonstrated that irregular or late bedtimes and shorter sleep duration may be risk factors for dental caries [8]. Although these studies did not focus on specific symptoms of sleep initiation or maintenance, together with our findings they suggest that irregular sleep patterns are relevant to children’s oral health.” (line 283)

Reviewer’s comment: The Kappa values between and within the examiners after calibration were reported to be > 0.76, indicating good consistency. To achieve the rigor of similar high-standard research in the [Knowledge Base] (PMID: 36122565), could you provide more detailed calibration results? For example, what are the Kappa values for the specific thresholds (ICDAS-II codes 5 and 6) used to define "generalized caries"? This is crucial for ensuring the measurement accuracy of the result variable.

Authors’ response: We thank the reviewer for the comment. The suggested change has been incorporated into the revised version of the manuscript.

Good intra- (Kappa coefficient >0.88) and inter-examiner (Kappa coefficient >0.76) agreement was observed for the assignment of ICDAS-II codes related to dental caries. Agreement for the assessment of extensive dental caries (ICDAS codes 5 and 6) was also evaluated, showing excellent intra-examiner (Kappa coefficient = 0.96) and inter-examiner (Kappa coefficient = 0.94) reliability.” (line 122)

Reviewer’s comment: It is an advantage that random sampling stratified by school type (public/private) was adopted. Given that your research hypothesis involves the key variable of socioeconomic status (SES), and the type of school (public vs. Private is usually a strong proxy metric for SES. In the subsequent analysis, do you plan to incorporate "school type" as a stratified variable or covariate into the model to more clearly dissect the possible complex relationship between SES, sleep habits and dental caries?

Authors’ response: We appreciate the reviewer’s comment. However, instead of including the type of school as a variable in the multiple regression analysis, we incorporated the sampling weights related to the type of school for each participant, as a multistage sampling design was used rather than simple random sampling. The results of this weighted adjusted model are presented in Table 2.

Reviewer 3 Report

Comments and Suggestions for Authors

Title: Extensive Dental Caries in Childhood: Association with Socio-economic Status, Dietary and Oral Hygiene Habits, and Sleep Disorders

This cross-sectional study investigated the prevalence of extensive dental caries (ICDAS-II codes 5–6) in Brazilian children aged 6–10 years and its association with socioeconomic factors, dietary habits, oral hygiene, and sleep disorders (SD). The methodology is generally sound, employing appropriate instruments (ICDAS-II and SDSC) and suitable statistical analysis (Poisson regression). The finding that Sleep Disorders (SD) are significantly associated with extensive caries, even after adjusting for traditional factors, is the strongest potential contribution.

However, the manuscript suffers from a lack of novelty regarding most findings and requires substantial strengthening of its Discussion and Literature Review to justify publication. The link between SD and caries, the paper's most novel aspect, is not fully explored due to data limitations.

  1. Lack of Novelty and Bibliographic Support

The associations found between extensive dental caries and lower parental education, and frequent consumption of sugary snacks/drinks, are widely established findings in the literature.

Action Required: The authors must critically revise the Introduction and Discussion to foreground the unique contribution of this study: the association between Sleep Disorders (SDSC Total Score) and the Severity of Caries (ICDAS-II 5-6). The manuscript must explicitly argue why this specific association is significant for public health and clinical management, rather than focusing on the well-known socioeconomic and dietary links.

Bibliographic Gap: The reference list (currently 23 items) appears limited for a study addressing a multifactorial disease. Authors must incorporate more robust and recent literature, specifically linking sleep issues to caries severity or advanced stages (not just general prevalence) to support their central hypothesis.

  1. Incomplete Testing of the Core Hypothesis

The proposed mechanism linking SD to caries is night-time sugar consumption, which combines with reduced salivary flow during sleep. However, the authors acknowledge that consumption of sugary foods/drinks immediately before bedtime was not assessed.

Action Required: This gap is a significant limitation. The Discussion must clearly state that while an association was found between SD severity and extensive caries, the proposed causal pathway (SD night-time sugar advanced caries) remains speculative and untested within this study's data. This limitation should be prominently featured.

  1. Methodological and Statistical Clarifications
  • Clustering Effect in Analysis: The sample size calculation included a correction factor of 1.2 for the cluster effect. The authors must confirm if this clustering (schools as sampling units was accounted for in the adjusted Poisson regression models to ensure the reported confidence intervals (95% CI) and -values are correct.
  • Collinearity Confirmation: The authors speculate that the non-significant finding for household income in the adjusted model is "possibly due to collinearity with the parental education variable".
  • Action Required: This speculation must be supported by evidence. Authors should run a formal test for collinearity (e.g., Variance Inflation Factor - VIF) and report the findings to justify the exclusion of household income as a significant factor.
  • Oral Hygiene (Toothbrushing): The lack of association between brushing frequency and extensive caries is noted. The discussion suggests the cause might be "inadequate brushing quality".
  • Action Required: The discussion should be improved by acknowledging that the measure used (frequency: times/day vs. times/day ) is insufficient to capture the true protective effect of oral hygiene, reinforcing that quality and technique are critical factors that were not assessed.

Variable Definition: Please clarify in Section 2.5 if the dependent variable, "presence of extensive dental caries" (ICDAS-II 5-6), refers to the presence of at least one such lesion per child (individual prevalence), across either primary or permanent teeth.

Table 2 Presentation: For completeness, the line item for the SDSC sum (total score) should be integrated fully within Table 2, including the Unadjusted PR/CI/p-value and Adjusted PR/CI/p-value, consistent with the other variables.

Generalizability: As the study was conducted in a single municipality (Diamantina, Minas Gerais, Brazil), the Discussion should include a more explicit statement on the limitations regarding the generalizability (external validity) of the findings.

Author Response

Reviewer #3

Reviewer’s comment: This cross-sectional study investigated the prevalence of extensive dental caries (ICDAS-II codes 5–6) in Brazilian children aged 6–10 years and its association with socioeconomic factors, dietary habits, oral hygiene, and sleep disorders (SD). The methodology is generally sound, employing appropriate instruments (ICDAS-II and SDSC) and suitable statistical analysis (Poisson regression). The finding that Sleep Disorders (SD) are significantly associated with extensive caries, even after adjusting for traditional factors, is the strongest potential contribution.

However, the manuscript suffers from a lack of novelty regarding most findings and requires substantial strengthening of its Discussion and Literature Review to justify publication. The link between SD and caries, the paper's most novel aspect, is not fully explored due to data limitations.

Authors’ response: We thank the reviewer for the comment. The Discussion section was revised to better address the association between sleep disorders and dental caries.

Reviewer’s comment: Lack of Novelty and Bibliographic Support

The associations found between extensive dental caries and lower parental education, and frequent consumption of sugary snacks/drinks, are widely established findings in the literature. Action Required: The authors must critically revise the Introduction and Discussion to foreground the unique contribution of this study: the association between Sleep Disorders (SDSC Total Score) and the Severity of Caries (ICDAS-II 5-6). The manuscript must explicitly argue why this specific association is significant for public health and clinical management, rather than focusing on the well-known socioeconomic and dietary links.

Authors’ response: We thank the reviewer for the valuable comment. The Introduction and Discussion sections were updated to strengthen the discussion of the association between sleep disorders and dental caries.

Reviewer’s comment: Bibliographic Gap: The reference list (currently 23 items) appears limited for a study addressing a multifactorial disease. Authors must incorporate more robust and recent literature, specifically linking sleep issues to caries severity or advanced stages (not just general prevalence) to support their central hypothesis.

Authors’ response:

Thank you for this comment. We have conducted an additional comprehensive review of the literature and incorporated several more recent and robust studies. These new references have been added to the manuscript to provide stronger empirical support for our central hypothesis. We believe that these additions enhance the theoretical grounding and contextual relevance of our study.

Reviewer’s comment: Incomplete Testing of the Core Hypothesis

The proposed mechanism linking SD to caries is night-time sugar consumption, which combines with reduced salivary flow during sleep. However, the authors acknowledge that consumption of sugary foods/drinks immediately before bedtime was not assessed. Action Required: This gap is a significant limitation. The Discussion must clearly state that while an association was found between SD severity and extensive caries, the proposed causal pathway (SD night-time sugar advanced caries) remains speculative and untested within this study's data. This limitation should be prominently featured.

Authors’ response: We thank the reviewer for this important comment. We acknowledge that the present study did not directly assess the mechanisms proposed in the literature linking sleep disorders to dental caries, such as nighttime sugar consumption combined with reduced salivary flow during sleep. Although this pathway has been suggested by previous studies, it could not be evaluated within the scope of the present investigation.

In particular, the consumption of sugary foods and beverages immediately before bedtime was not assessed, which represents a limitation of the study. It has now been clearly stated and highlighted in the Discussion to ensure appropriate interpretation of the observed association between sleep disorder severity and extensive dental caries.

“Although salivary flow was not assessed in the present study, the literature suggests that nighttime sugar consumption coincides with a natural reduction in salivary flow during sleep, which is an important protective factor against dental caries [6]. However, this pathway was not evaluated in this study. Specifically, the consumption of sugary foods or beverages immediately before bedtime was not assessed, which represents a limitation of the present study. Given the cross-sectional design and the nature of the data collected, it was not possible to directly observe or measure these behaviors.” (line 311)

Reviewer’s comment: Methodological and Statistical Clarifications

Clustering Effect in Analysis: The sample size calculation included a correction factor of 1.2 for the cluster effect. The authors must confirm if this clustering (schools as sampling units was accounted for in the adjusted Poisson regression models to ensure the reported confidence intervals (95% CI) and -values are correct.

Authors’ response: Following the reviewer’s suggestion, we performed a new multiple regression analysis incorporating sampling weights related to the type of school for each participant, as a multistage sampling design was used rather than simple random sampling. The results of this weighted adjusted model are presented in Table 2. In addition, we revised the Statistical Analysis section to clarify how clustering by type of school was addressed through the use of sampling weights.

“All analyses were conducted taking into account the weights of the type of school of each participant, since multistage sampling method was used instead of simple random sampling.” (line 181)

Reviewer’s comment: Collinearity Confirmation: The authors speculate that the non-significant finding for household income in the adjusted model is "possibly due to collinearity with the parental education variable". Action Required: This speculation must be supported by evidence. Authors should run a formal test for collinearity (e.g., Variance Inflation Factor - VIF) and report the findings to justify the exclusion of household income as a significant factor.

Authors’ response: Following the reviewer’s previous suggestion, we revised the multiple regression analysis by incorporating sampling weights related to the type of school, in order to account for the multistage sampling design. In this weighted adjusted model, household income remained statistically significant. The revised results are presented in Table 2, and the previous discussion about collinearity was removed from the Discussion section.

In the present study, even after statistical adjustment, the association between household income and dental caries remained significant.” (line 265)

Reviewer’s comment: Oral Hygiene (Toothbrushing): The lack of association between brushing frequency and extensive caries is noted. The discussion suggests the cause might be "inadequate brushing quality". Action Required: The discussion should be improved by acknowledging that the measure used (frequency: times/day vs. times/day) is insufficient to capture the true protective effect of oral hygiene, reinforcing that quality and technique are critical factors that were not assessed.

Authors’ response: We thank the reviewer for this important comment. We agree that toothbrushing frequency alone is insufficient to capture the true protective effect of oral hygiene. In the revised Discussion, we explicitly acknowledge that the measure used in this study (daily toothbrushing frequency) does not account for brushing quality, technique, duration, or supervision, which are critical factors in caries prevention. This limitation has now been clearly stated in the Discussion to ensure appropriate interpretation of the lack of association observed.

“However, this lack of association may be explained by the limitations of the measure used, as daily toothbrushing frequency alone is insufficient to capture the true protective effect of oral hygiene. Important aspects such as brushing quality, technique, duration, and caregiver supervision were not assessed in this study and may play a critical role in caries prevention.” (line 325)

Reviewer’s comment: Variable Definition: Please clarify in Section 2.5 if the dependent variable, "presence of extensive dental caries" (ICDAS-II 5-6), refers to the presence of at least one such lesion per child (individual prevalence), across either primary or permanent teeth.

Authors’ response: We thank the reviewer for the comment. The following sentence was included in the Materials and Methods section:

“Children presenting at least one dental caries lesion classified as ICDAS-II code 5 or 6, in either primary or permanent teeth, were classified as having the presence of extensive dental caries.” (line 163)

Reviewer’s comment: Table 2 Presentation: For completeness, the line item for the SDSC sum (total score) should be integrated fully within Table 2, including the Unadjusted PR/CI/p-value and Adjusted PR/CI/p-value, consistent with the other variables.

Authors’ response: We thank the reviewer for this suggestion. Table 2 has been revised and adjusted.

Reviewer’s comment: Generalizability: As the study was conducted in a single municipality (Diamantina, Minas Gerais, Brazil), the Discussion should include a more explicit statement on the limitations regarding the generalizability (external validity) of the findings.

Authors’ response: We thank the reviewer for this important comment. We have revised the Discussion section to address the limitations related to the study’s external validity, acknowledging that the findings are derived from a single municipality and may not be fully generalizable to other populations or settings.

“Finally, it is important to note that, as the data of the present study were obtained from a single municipality in southeastern Brazil, the findings may have limited external validity and should be interpreted with caution when extrapolated to other settings or contexts. Regional differences in socioeconomic conditions, access to dental care, and health-related behaviors may influence the occurrence of sleep disorders and dental caries. Therefore, further studies conducted in diverse settings are needed to confirm and expand upon these findings.” (line 342)

Round 2

Reviewer 3 Report

Comments and Suggestions for Authors

Dear authors,

You have made many improvements to the article. I recommend publishing it.